# Microfluidic Obstacle Arrays Induce Large Reversible Shape Change in Red Blood Cells

**DOI:** 10.3390/mi12070783

**Published:** 2021-06-30

**Authors:** David W. Inglis, Robert E. Nordon, Jason P. Beech, Gary Rosengarten

**Affiliations:** 1School of Engineering, Macquarie University, Sydney, NSW 2109, Australia; 2Graduate School of Biomedical Engineering, University of New South Wales Sydney, Sydney, NSW 2052, Australia; r.nordon@unsw.edu.au; 3Division of Solid State Physics and NanoLund, Physics Department, Lund University, P.O. Box 118, 22100 Lund, Sweden; jason.beech@ftf.lth.se; 4School of Engineering, RMIT University, Melbourne, VIC 3053, Australia; gary.rosengarten@rmit.edu.au

**Keywords:** shear, erythrocyte, morphology, microfluidic, eterministic lateral displacement (DLD)

## Abstract

Red blood cell (RBC) shape change under static and dynamic shear stress has been a source of interest for at least 50 years. High-speed time-lapse microscopy was used to observe the rate of deformation and relaxation when RBCs are subjected to periodic shear stress and deformation forces as they pass through an obstacle. We show that red blood cells are reversibly deformed and take on characteristic shapes not previously seen in physiological buffers when the maximum shear stress was between 2.2 and 25 Pa (strain rate 2200 to 25,000 s^−1^). We quantify the rates of RBC deformation and recovery using Kaplan–Meier survival analysis. The time to deformation decreased from 320 to 23 milliseconds with increasing flow rates, but the distance traveled before deformation changed little. Shape recovery, a measure of degree of deformation, takes tens of milliseconds at the lowest flow rates and reached saturation at 2.4 s at a shear stress of 11.2 Pa indicating a maximum degree of deformation was reached. The rates and types of deformation have relevance in red blood cell disorders and in blood cell behavior in microfluidic devices.

## 1. Introduction

The red blood cell’s unique mechanical properties, a tough, pliable membrane and lack of internal filaments and tubules, enable it to survive repeated extreme deformations including passage through capillaries. There has been a significant amount of work on red blood cell shape and behavior under shear stress [1,2,3,4]. It is well known that red blood cells in cylindrical capillaries deform into symmetric and non-axisymmetric parachutes. Tsukada et al. [5] showed elongated parachutes in glass and in vivo capillaries with shear stress up to 1 Pa. Dodson et al. [6], Basu et al. [7] and Abkarian et al. [8] observed tank treading, tumbling, and swinging at comparatively low shear stresses of <560, <30, <5 mPa, respectively. Beech et al. [9] showed red blood cell elongation at constrictions in post arrays at shear stress up to 8400 mPa.

The effect of chemical forces on RBC morphology has been the subject of scientific enquiry for longer still. Hypotonic solutions are well known to cause cells to form swollen but invaginated shapes—stomatocytes. Hypertonic solutions lead to spiculated shapes—echinocytes. The continuum of these and normal shapes was described by Marcel Bessis in 1972 [10] and is known as the stomatoycte-discocyte-echinocyte (SDE) sequence. Normal red cells have been observed to reversibly form shapes outside of the SDE sequence through chemical treatments. Non-axisymmetric (NAS) discocytes are discocytes with an uneven rim thickness. Triangular stomatocytes have a deep triangular invagination. Knizocytes are triconcave cells that can be found in healthy newborns. Additional shapes are seen in various pathologies. Knizo-echinocytes—triconcave cells with 3 or 4 bulges on each ridge—have been predicted.

Chemicals that alter the membrane mechanics give insights into stable shapes, but deformations induced by flow in microchannels occur in milliseconds. Many methods exist to measure deformability and mechanical properties of cells. Micropipette aspiration and, more recently, optical tweezers have produced the most straightforward data and have informed sophisticated numerical modeling of the plasma membrane [2,11,12]. RBC transit through variously shaped microchannels has also become a tool for quantifying cell properties [13].

In contrast to other work, where cells are exposed to a single stress event, our work examines RBCs in flowing pulsatile shear stress where cells rotate between each stress event. The flow speed determines the frequency of shear stress pulsation; here it is between 560 Hz and 58 kHz. This unique structure is the result of flow through a particle separation system called deterministic lateral displacement (DLD) [14]. Deterministic lateral displacement is used for blood processing [15,16]. Studying RBCs in these systems is relevant for understanding and optimizing cell separation, and may give insight into RBC biophysics.

In this paper, we show that RBCs flowing through DLD-style arrays with a peak stress of 2.2 to 25 Pa (2200 to 25,000 s^−1^), reversibly take on shapes not previously described in studies of shear stress. Figure 1a shows an example of deformation as a red blood cell enters an array of cylindrical obstacles having a 7 μm gap. Figure 1b shows a cell relaxing or recovering normal morphology upon exiting the obstacle region. The obstacle array is titled by a small amount (1–2°) with respect to the mean fluid flow direction. This tilt makes the hydrodynamic environment substantially different than capillary Poiseuille flow. In a cylindrical capillary, the labile RBC moves away from the high-shear, no-slip boundary. In the tilted pillar array, the cell cannot avoid repeated high-shear contact with surfaces.

## 2. Experimental Details

A microfluidic device was designed to allow investigation of RBCs flowing through obstacle arrays. Arrays have a 7 μm gap and 18 μm diameter posts. Arrays are 1225 μm wide (49 rows). The array tilt was either 1/30 or 1/50. The array length was 3.75 and 6.25 mm, respectively. The fluidic channel is 32 μm deep. Devices were fabrication by a previously described soft lithography method [17].

Venous blood was harvested from human donors under ethics approval in EDTA coated tubes and used within 24 h. Peripheral blood mononuclear cells were removed by centrifugation at 600× g for 5 min. The resulting packed red cells were diluted in AutoMACs buffer (Miltenyi Biotech, Bergisch Gladbach, Germany), a phosphate buffered saline containing EDTA anticoagulant and BSA (bovine serum albumin). Devices were placed on an inverted microscope fitted with a Phantom V7.3 high-speed camera (Vision Research Inc., Charlottetown, Canada) and 40× objective giving a field of view on the digital image of 359 × 269 μm^2^ (0.57 μm/pxl). The microscope stage was stationary during video capture. Cell tracking was performed in ImageJ with the MtrackJ plugin.

The device was connected to a tube of diluted red blood cells using a 12 cm length of 0.6 mm ID silicone tubing. The tube was pressurized using a syringe to between 0.1 and 1.5 psi (3.5 to 28 kPa) and the flow of cells was observed. The cells’ speed and the time/location of shape change events at the entrance or exit of the array were recorded.

Flow speed was determined by measuring the speed of red cells in regions containing no posts either upstream or downstream of the array. In these locations, we assume the red cells behave as Lagrangian fluid tracers. The fastest 10% of 20 measured cell speeds is taken as the maximum fluid speed at that location.

To estimate the shear rate and stress that cells are exposed to, we created a 3-dimensional computational fluid dynamics model (COMSOL) of a single pair of pillars, see Appendix A for details of the model. Shear stress is the viscosity of water (0.001 Pa s) times shear rate. Running buffer is phosphate buffered saline with ~1% bovine serum albumin. This solution is expected to have a viscosity that is not more than 10% larger than that of water [18]. From the model, we can say that the peak shear rate in the array, which occurs on the post surfaces at the point of greatest constriction, is 2605 s^−1^ when the speed in the post-free region is 1 mm/s, the peak shear stress is 2.6 Pa and Reynolds number between the posts at this speed is 0.037. All these numbers scale linearly with fluid velocity. Reynolds number is defined as the average velocity times the minimum gap distance over kinematic viscosity.

To monitor cell relaxation within the camera field of view, the flow was stopped while observing the exit of the array. Flow was stopped by clamping the silicone tubing either immediately upstream or downstream of the device. Using the high-speed camera, we captured the entire clamping and deceleration process, giving a precise measure of speed before, during and after clamping. Fluid and cells take between 50 and 300 ms to decelerate during and after the clamp is applied, depending on initial speed and the fluid volume that must flow to equilibrate pressures. To provide a reasonable number of cells which all have a similar and clearly defined flow/shear history, only cells that exited the array within +/−100 ms of the beginning of the deceleration were recorded. This time window was chosen to be significantly less than the relaxation time. Cell density in the array varied due to speed-dependent settling in inlet reservoir and tubing and ranged from 18 to 120 million cells per mL, a 290- to 47-fold dilution from whole blood, thus we use the viscosity of water, 0.001 Pa⋅s.

In this work, we measured time to deformation and time to relaxation for individual cells once the high shear events begin or end. In contrast to prior work [19], it is the entrance and exit from the pillar array that initiated and terminated the high shear events, and not the stopping or starting of the fluid flow. Flow is stopped to keep the cell in the camera’s field of view. The shape of each cell is assessed at the centerline between successive columns of obstacles where it moves slowest. The shape of cells while in contact with pillars is not considered. A cell is considered to be deformed when it is clearly neither a discocyte or stage I stomatocyte. In many cases, it was impossible to differentiate between discocyte and stage I stomatocyte morphologies, so the two classes have been grouped together. A stage 1 stomatocyte is a discocyte that is concave on only 1 side, and flat or convex on the other side.

Other work on RBC deformation in microchannels showed it is possible to define a deformation ratio [13]. A similar definition here is difficult because the deformations take such a wide variety of forms including shapes that are highly deformed but not extended (Figure 1a at 32 ms, and Figure 2 other).

The time to relaxation is defined as the interval between exiting the array and returning to discocyte or stage I stomatocyte morphology. The empirical probability of these events (Kaplan–Meier estimator [20]) was calculated from these time intervals using the MATLAB statistical toolbox. Rates of change were calculated by weighted linear regression on log empirical probability.

## 3. Results and Discussion

### 3.1. Cell Shapes and Size

We observed that at peak shear rates that are comparable to the low physiological range, cells retain their shape throughout the arrays. Under these conditions, red blood cells tumble, and bend by up to 90° as they collide with obstacles, but generally recover their discocyte shape between rows of posts and show very little accumulation of shape change upon traversing 100 gaps. At the lowest pressure that was applied (0.1 psi), cells had a maximum shear stress of 1.4 Pa. For comparison, peak physiological shear stresses range from 1 to 10 Pa with pulsation frequencies that equal the heart rate [21]. Pulsation frequencies in the microfluidic device are given by the cell velocity divided by the array pitch (25 μm) and range from 32 to 320 Hz.

At higher flow speeds, shear forces deform the cells and cells do not recover their discocyte shapes between gaps. We observed no cell lysis or membrane rupture, consistent with multiple examples showing that lysis occurs at shear stresses more than 100 Pa [22,23]. We observed no stage II, III or IV stomatocytes and no echinocytes. Figure 2 shows examples of some of the shapes observed in and immediately after the array.

Particularly at the highest speeds where cells are highly folded, the cells have an increased surface area to volume ratio which can occur in two ways. Volume changes can occur via the transport of water through the membrane. This is known to occur due to osmotic pressure but the reverse (transport of water out of the cell due to compressive forces) has not been reported. A consequence of a decrease in the water content of the RBC is an increase in the hemoglobin concentration and a resulting increase in the internal viscosity of the cell which is known to affect RBC dynamics [24]. A decrease in the water content of the RBC will also increase the surface area to volume ratio. Namvar et al. showed that changes to the surface area to volume ratio have a larger effect on deformability than changes to membrane elasticity [25].

Surface area changes due to the application of forces have been reported. Park et al. propose that there is an excess of lipid bilayer in the RBC under physiological conditions and that the spectrin network is responsible here for the elastic properties [26]. Spectrin is an entropic spring with a contour length of 170 nm and the pore size in the triangular spectrin network is 90 nm. This means that the RBC area can potentially increase until either: 1. The spectrin reaches full extension or 2. The excess lipid is depleted. Both 1 and 2 would lead to a large change in the RBCs’ elastic properties and a change to the surface area to volume ratio. As described below, both the deformation and recovery that we observe are nonlinear and likely have contributions from both an increase in the cell membrane area, and a decrease in cell volume. This contrasts with Lim and Wortis et al. who simulated some of these and other RBC shapes through an increased internal volume [27].

### 3.2. Deformation Rates

The variation in time to deformation and recovery led us to analyze this data using statistical lifetime analysis [20]. Time to deformation for each cell was used in empirical cumulative probability distributions where a deformation rate is calculated by weighted linear regression of logarithm of survival/recovery. The rate is the inverse of a time constant in a simple exponential decay, and is used to calculate when most cells have deformed or relaxed (e^−3^ = 0.05, so 95% of cells have transformed when *t* = 3/rate).

Figure 3 shows the empirical probability of cells avoiding deformations upon entering the array. The top row of graphs expresses the probability with respect to time, while the bottom row shows it as a function of penetration into the array. Three different flow speeds are compared, with peak shear stresses of 1.86, 10.5 and 21.2 Pa. The last panel in each row shows how the rate of deformation per second and per gap traversed change with flow rate (peak shear). We observed that the rate of deformation per second increases with shear rate. At a peak stress of 1.86 Pa, the rate is 9.25 s^−1^, meaning that 95% of cells will deform in 320 ms. At the faster flow rate (peak stress of 21.2 Pa), the rate is 132 s^−1^, meaning that 95% of cells will deform in 23 ms.

In contrast with the clear speed dependence of the time-rate of deformation (Figure 3, top right), the rate of deformation per gap traversed has much less dependence on shear (Figure 3, bottom right). The rate goes from 0.175 to 0.22 gap^−1^, meaning that 95% of cells deform in 17 gaps at the lowest speed and 14 gaps at the highest speed. Increased flow speeds do not therefore drastically increase the probability that a post collision will result in deformations. This translates into a speed-independent deformation zone at the beginning of the array in which cells are likely to have a normal morphology. Only after this zone do they take on the deformed shapes. We define the deformation zone as the depth into the device where the deformation probability is greater than 0.5. This zone is 3 to 4 obstacles deep. Thus, we conclude that it is the number of gaps traversed and not the peak shear rate that ultimately determines when cells deform.

While true on average, many cells demonstrate behavior at either extreme. Some cells experience large and semi-permanent deformations from a single obstacle interaction while flowing slower than the average, and other cells accumulate small deformations as they flow quickly through the middle of successive gaps.

### 3.3. Relaxation Rates

Upon exiting the array, the maximum shear rate drops quickly because of two effects. The minimum channel dimension increases from 7 μm—the gap, to 32 μm—the depth. The maximum fluid speed decreases by a factor of 4 because of the increased channel volume after the array. The maximum shear that a cell could experience after exiting the array is then (4 × 32/7=) 18 times lower than it was in the array.

A further reduction in shear occurs as we stop the flow to enable imaging for 2 s after exit. Figure 4 shows two 100 μs exposures—the first, as cells exit the obstacle array at nearly full speed and the second, taken 2 s later after the cells have slowed, flowed backward a small amount, and stopped. The in-focus cells have largely recovered a normal morphology.

Figure 5a–e show the probability of relaxation to normal morphology for five flow speeds and shear rates. The solid black line on each is a single exponential fit where the fitting parameter is the relaxation rate. Figure 5f plots the relaxation rates from each experiment versus peak shear rate. At low flow and shear rates, most of the cells exit the array as discocytes or stage I stomatocytes that are slightly bent, but quickly relax to more normal shapes. At higher shear rates, cells exit the array in a variety of shapes, such as those seen in Figure 1 with knizocytes being common. They then progress toward non-axisymmetric (NAS) discocytes, slight parachutes or strongly bent discocytes before relaxing to discocytes or possibly stage 1 stomatocytes.

Some cells exit with normal morphologies: 28% at the lowest speed, 5% at a peak stress of 5.44 Pa, and 0% at higher speeds/stresses. Cells exiting with normal morphologies are not included in the calculation of rates. The relaxation rate versus peak shear decreases steeply on a logarithmic scale from 35 s^−1^ at the lowest shear stress to about 1.25 s^−1^ for the faster flow experiments. Considering the cells that did exit as normal, we can say that 95% of cells are normal after 0.031, 0.82, 1.5, 2.4, and 2.3 s in each of the 5 experiments from slowest to fastest flow/shear (Figure 5f).

The time to relaxation should depend on the degree of deformation from which the cells need to recover with larger deformation requiring longer times. As the time to relaxation saturates at a peak stress of approximately 10 Pa, we conclude that the degree of deformation also saturates under these conditions. This is consistent with the idea that the spectrin reaches a maximum degree of extension and/or the excess lipid is fully integrated into a single, taut, bilayer. We observe that the highly folded shapes disappear in a few seconds, while the final relaxation from NAS discocyte to normal RBC takes 10 to 20 s.

We now compare our relaxation results with those reported in the literature. Evans [28] observed shape recovery of an RBC ejected from a 4 μm inner diameter pipette. Full recovery took about 1 s with a time constant of 0.3 s or a rate of 3.3 s^−1^. Under extensional deformation (an increase in length to width ratio from 1 to 2), he observed slightly faster recovery and a time constant of 0.1 s or a rate of 10 s^−1^. Englehard et al. [29] observed the recovery of cells that had been very slightly elongated by an electric field. They constructed a model consisting of two damped springs (a fast and a slow exponential) with time constants of 0.16 and 0.9 s (rates of 6.25 and 1.1 s^−1^). For small deformations (low shear) we see a rate of relaxation that exceeds those previously measured. This is partially explained because we over-estimate the rate by counting stomatocyte I cells as “normal”. For higher shear and higher deformations, our relaxation rate is close to that measured for an unfolding RBC by Evans [28] (3.3 s^−1^) and the slow rate measured by Engelhardt et al. [29] (1.1 s^−1^). The comparison is complicated as each cell experiences a slightly different path through the array, and often, very different degrees of deformation with a combination of shear and extensional forces.

## 4. Conclusions

We have observed red blood cells flowing through an obstacle array where obstacle interactions cannot be avoided. We have found that when the maximum shear stress exceeds 1.5 Pa, the cells are rapidly and significantly deformed. In between the gaps and immediately after the array, cells take shapes like those predicted by models of altered membranes. In contrast with such models, cells in the array do not have an increased internal volume. These deformations are reversible and short-lived, taking less than 2.4 s to disappear. We observe that cell deformation occurs faster with faster flow rates, but the rate is more influenced by the number of shear spike events (gaps traversed). We see little change in the number of gaps traversed before deformation across an order of magnitude in shear. Recovery rates also indicate that a maximum degree of deformation is reached. Further studies of recovery rates may indicate whether they are limited by restructuring of the bilayer, of relaxation of the spectrin network, by viscosity of the membrane or the cytoplasm, or by transport of water through the membrane, and would elucidate what governs deformation dynamics.

Studying the dynamics of cells under shear stress has historically provided insights into fundamental red cell mechanics and these findings show a new phenomena. Furthermore, the work is important for designing new microfluidic separation systems that rely on size, such as deterministic lateral displacement [30]. The changes in cell mechanics that result in the characteristic shapes are unclear, and further work is needed to see the precise effect that shape change behavior has on deterministic lateral displacement efficiency.

## Figures and Tables

**Figure 1 micromachines-12-00783-f001:**
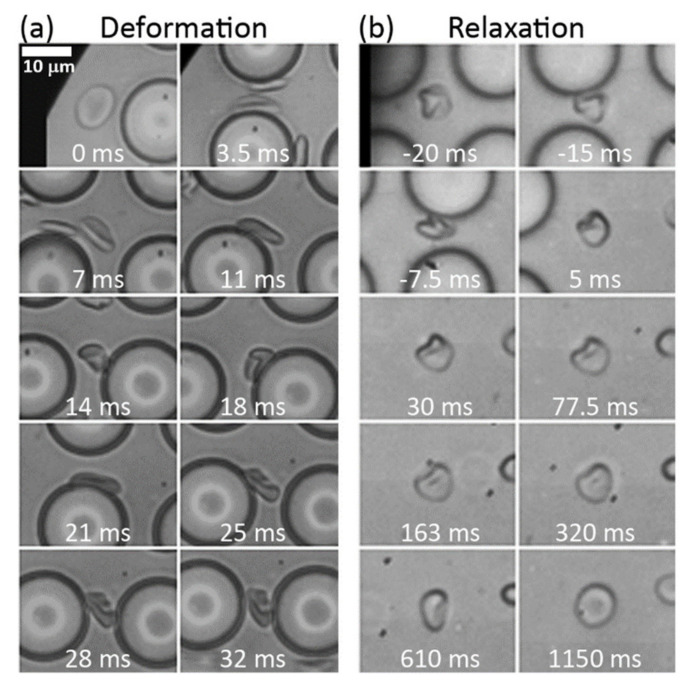
Series of images showing deformation and relaxation. Flow direction is left to right: (**a**) A red blood cell enters the array at *t* = 0. First major deformation is at 14 ms. Peak shear stress in the array is 13.6 Pa and the R_e_ in the minimum gap is 0.19. See Appendix A in Appendix A; (**b**) A red blood cell exits the array at *t* = 0 and takes approximately 1 s to recover. Frames are grabbed on an exponential time scale. Peak shear stress in the array is 5.5 Pa, see Appendix A.

**Figure 2 micromachines-12-00783-f002:**
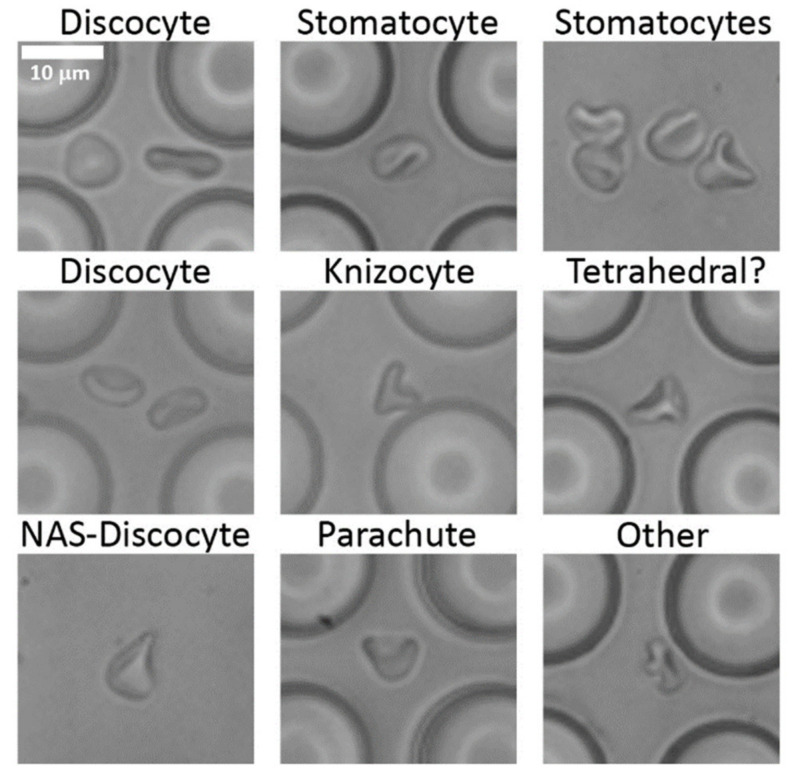
Examples of red blood cells observed in the device; NAS is non-axisymmetric. The flow direction is from left to right.

**Figure 3 micromachines-12-00783-f003:**
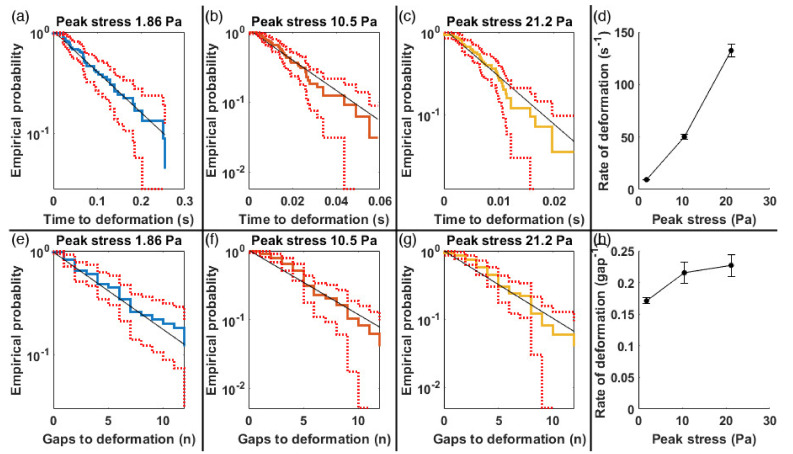
Empirical cumulative probability distribution for avoiding deformation upon entering an obstacle array at three flow speeds: (**a**–**c**) show the probability of deformation versus time, while (**e**–**g**) shows the probability of deformation versus the number of gaps traversed. Dashed red lines show 95% confidence bounds, 50 cells were analyzed in each case. The time to cell deformation was approximated by an exponential random process whose rate was estimated by weighted linear regression performed on log of empirical probability (thin black line); (**d**,**h**) show the rate of deformation per second and per gap versus peak shear stress.

**Figure 4 micromachines-12-00783-f004:**
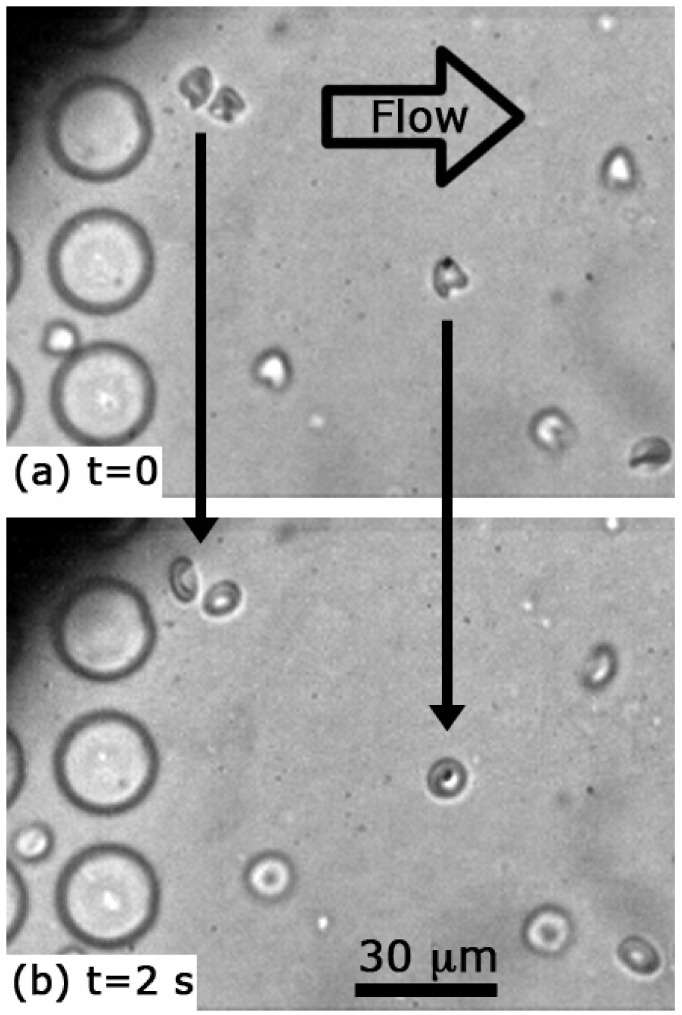
Comparison of cells just after leaving the obstacle array with the same cells, 2.07 s later (**a**,**b**). Peak shear stress in the array is 12.1 Pa. The cells in the top panel are decelerating from a peak speed in the open region of 5.3 mm/s and the velocity in the bottom panel is 0.

**Figure 5 micromachines-12-00783-f005:**
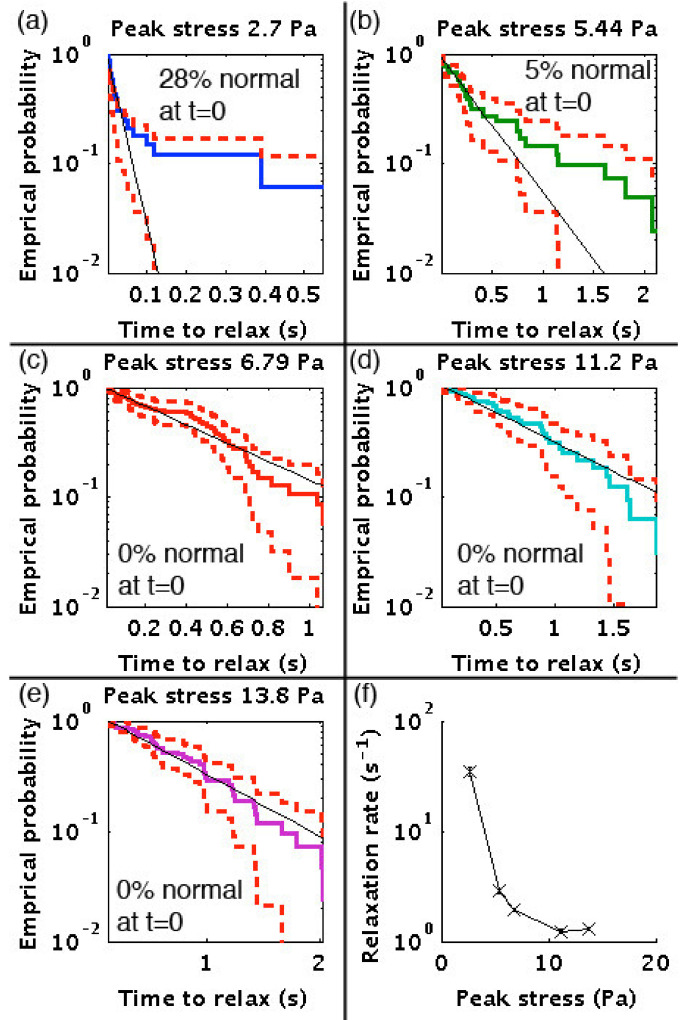
Empirical probability of the time to recovery for cells exiting the array at increasing flow rates (**a**) through (**e**). The solid black line is the weighted linear regression of log probability. The percentage of cells that exited the array with normal morphologies is given in each plot. (**f**) shows the relaxation rates versus peak shear stress in the array. The number of cells in each experiment is 46, 45, 47, 32 and 43.

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
