# Peer review of "Microfluidic Obstacle Arrays Induce Large Reversible Shape Change in Red Blood Cells"

_micromachines, 2021, doi:10.3390/mi12070783_

Round 1
Reviewer 1 Report
The manuscript described an experimental set-up for RBCs deformability measurements and RBCs shape discussion. Although written in a simple and clear way the contents/results of the paper are poor, looking to the ones already published in the literature.
In my opinion, some control experiments are missing and comparison with other literature results. Some of my comments:
Since the authors refer that “the rates and types of deformation have relevance in red blood cells disorders and in blood cell behavior in microfluidic devices”, the values of shear rates and strain used should be compared to the ones found in in vivo and discuss.
The values of deformability should be compared with the ones found in the literature. The authors refer that the RBCs are exposed to pulsatile shear stress. However in in vivo they still under a continuous shear flow. Thus, and as other authors have already performed, a hyperbolic microfluidic shaped device will be much more accurate to measure deformability and recovery rates.
Scale bar the figure 1 and 2 is missing. Figure2 something wrong with the legend.
The device has a high depth, thus the possible rotation of the RBCs (in each on-axis) should be considered in order to not influence the measurements.
What haematocrit was used? Does it proportionate reliable results and representative of the real behavior of RBCs?
How the paper is organized and discuss, seems that the authors only want to check the behavior of the cells in the device but, and the final application of that? They will use that results for what?
Author Response
Reviewer #1:
1. the values of shear rates and strain used should be compared to the ones found in in vivo and discuss.
We have revised section 3.1 to include a comparison to physiological shear.
2. The values of deformability should be compared with the ones found in the literature. The authors refer that the RBCs are exposed to pulsatile shear stress. However, in in vivo they still under a continuous shear flow. Thus, and as other authors have already performed, a hyperbolic microfluidic shaped device will be much more accurate to measure deformability and recovery rates.
We are not able to quantify deformability. More importantly, the goal of this research was not to measure a fundamental RBC property, nor to investigate the effects of parabolic shear on RBCs. As the reviewer points out, this work has already been done.
Our work shows a new phenomenon that has not been described before. It shows a phenomenon that is specifically caused by a unique pulsatile stress that is found in certain types of microfluidic blood separation devices. We have added two recent references showing blood separation using pillar arrays: Introduction, references 15 and 16.
3. Scale bar the figure 1 and 2 is missing. Figure2 something wrong with the legend.
We have added scale bars to Figures 1 and 2.
The text box surrounding the figure was incorrectly sized and was obscuring the caption. Thank you for picking this up. We have corrected the error.
4. The device has a high depth, thus the possible rotation of the RBCs (in each on-axis) should be considered in order to not influence the measurements.
The geometry of the device represents a real particle separation array, which must have significant depth to allow for reasonable throughput. In such a real device, 3 degrees of rotational freedom is a reality and does not detract from the measurement. If we were trying to make quantitative measurements of RBC deformability we would use a different device.
5. What haematocrit was used? Does it proportionate reliable results and representative of the real behavior of RBCs?
The cells are run at low number density and negligible haematocrit to avoid cell-cell interactions. The effect of such interactions would likely complicate the flow, the behavior of the cells and the analysis of their individual dynamics.
By real behavior, we believe that the reviewer is referring to behavior in vivo. This seems irrelevant as the shear conditions in the array are not designed to mimic in vivo conditions.
6. How the paper is organized and discuss, seems that the authors only want to check the behavior of the cells in the device but, and the final application of that? They will use that results for what?
The goal of this paper is to present an observation of RBC dynamics in pillar arrays. The work sheds light on how RBCs can be separated from other cells in Deterministic Lateral Displacement Arrays. This application has been expanded upon in the introduction and was noted in the final paragraph of the conclusion. Furthermore, we believe that this work fits very well with the theme of this special issue.
Reviewer 2 Report
The article “Microfluidic obstacle arrays induce large reversible shape change in red blood cells” studied RBC deformation and shape recovery under different shear stress flows using DLD setup. The authors have observed that with increasing shear stress, the shape recovery takes few tens of milliseconds to as high as few seconds. RBCs, while passing through DLD devices, adapt many shapes such as stomatocytes, trilobes, parachutes etc. The major finding of the manuscript is that shape recovery of RBC depends on the shear stress applied. The recovery time increases with the shear stress which is related to increased RBC deformations with increased shear stress. The study is interesting and relevant to the field. I have some following questions.
- In Fig. 5, empirical probabilities are shown at different peak shear stress. It would be better to measure and plot relaxation times or shape recovery times for different peak shear stress directly instead of empirical probabilities.
- Few tens of milli seconds of relaxation time probably result from RBC elasticity. Relaxation time in seconds can only be explained with spectrin remodeling. It has also been shown recently that RBC that enters small constrictions also display longer relaxation times (see here: doi:10.1038/srep43134). Is it possible to map different shapes to different relaxation times? For example, statement like “certain shapes require spectrin remodeling and while other shapes don’t” can be made?
- It would also be interesting to perform experiments on stiffened RBCs as many diseases make RBC stiffer to see how stiffness affect the shapes as well as the shape recovery.
Author Response
Reviewer #2:
1. In Fig. 5, empirical probabilities are shown at different peak shear stress. It would be better to measure and plot relaxation times or shape recovery times for different peak shear stress directly instead of empirical probabilities.
We believe that Figure 5 panel (f), does that. Figure 5 (a) to (e) allow us to fit a single exponential for each of 5 different shear stresses. The time constant for the exponential fit is the relaxation rate. This relaxation rate is plotted against peak shear stress in figure 5 (f).
We have made changes to the description of Figure 5 in the main text to help clarify this.
2. Is it possible to map different shapes to different relaxation times? For example, statement like “certain shapes require spectrin remodeling and while other shapes don’t” can be made?
Regrettably we have not been able to recover the original videos as they are from a number of years ago. We have 5 representative videos of cells exiting the array and they show that the highly folded shapes disappear in a few seconds, while the final relaxation from NAS discocyte to normal RBC take 10 to 20 seconds.
We have added a statement to this effect in section 3.3.
3. It would also be interesting to perform experiments on stiffened RBCs as many diseases make RBC stiffer to see how stiffness affect the shapes as well as the shape recovery.
Again, the reviewer makes a very useful suggestion; however, the infrastructure to repeat it is not currently available.

Round 2
Reviewer 1 Report
The paper has been updated and the suggestions addressed to the new version of the manuscript.